# Citius, Altius, Fortius; Is It Enough to Achieve Success in Basketball?

**DOI:** 10.3390/ijerph17207355

**Published:** 2020-10-09

**Authors:** Javier García-Rubio, Daniel Carreras, Sebastian Feu, Antonio Antunez, Sergio J. Ibáñez

**Affiliations:** 1Training Optimization and Sports Performance Research Group (GOERD), Sport Science Faculty, University of Extremadura, 10005 Caceres, Spain; danielcarreras1995@gmail.com (D.C.); Antunez@unex.es (A.A.); sibanez@unex.es (S.J.I.); 2Facultad de Educación, Universidad Autónoma de Chile, Santiago 7500912, Chile; 3Facultad de Educación, University of Extremadura, 60001 Badajoz, Spain; sfeu@unex.es

**Keywords:** physical conditioning, performance, basketball

## Abstract

The NBA Draft Combine includes a series of standardized measurements and drills that provide NBA teams with an opportunity to evaluate players. The purpose of this research was to identify the Combine tests that explain draft position and future performance in the NBA rookie season. Variables were selected from the previous categories of anthropometric measurements and strength and agility tests. A regression analysis was carried out. Combine variables, anthropometric and agility/strength variables were analyzed to explore their effect on draft position. Moreover, correlation analyses were performed to identify relationships among: (i) Combine anthropometric and strength and agility measures and game performance through game related statistics; and (ii) the draft position and game performance using Pearson’s correlation coefficients. Results show that the Combine test does not predict draft position, with the exception of hand width and height in frontcourt players, and standard vertical jump and running vertical jump. Future performance indicators were explained by several Combine tests in all players.

## 1. Introduction

It has been stated that several physical capacities or athletics abilities are related with high level performance and can influence players′ game performance, such as morphological characteristics, physical fitness or technical and tactical skills [1,2,3]. In fact, athletic measurement has been widely used with future sport performance success purposes [4]. However, research on National Basketball Association (NBA) and National Football League (NFL) players has shown that higher picks in the draft will not always have successful careers [5,6]. Moreover, research has pointed out that the best teams recruit young players according to their chronological age [7], meaning that more mature players have better opportunities of being scouted. In this respect, it has been hypothesized that, perhaps, on-court performance, measured from game-related statistics (points, rebounds…), is a better indicator of future performance than fitness measurements with physical tests (anthropometric and strength and agility) in the Combine [8]. 

The fitness evaluation of basketball players is carried out through generic or specific tests for sports both with laboratory and field tests. The literature review reveals the paucity of specific tests to assess the physical qualities of basketball players [9]. For an ecological evaluation and greater application and transfer to the real context of the competition, it is necessary to use specific and field tests [10], which involve the identification and motivation of the players.

Athletic ability is positively associated with future success in basketball, especially morphological features [11]. In the NFL, several athletics measurements such as power [12], sprint time [13] or vertical jump height in collegiate players [14] have been associated with success. Moreover, in hockey it has been stated that players’ speed, strength, power and body composition are critical components for play [15,16]. In those sports where physical contact is allowed and necessary to succeed for some playing positions, such as in football or hockey, speed is fundamental. In basketball, the playing field is smaller, so speed is not as crucial as in football. However, in both sports the context is continuously changing and unpredictable, so other skills are also important to succeed. Specifically, in the basketball Combine, drafted players outperformed undrafted players in height, wingspan, vertical jump height and reach, line agility and three-quarter sprint test in all five playing positions. Specifically, drafted guards achieved better results in leg power and height and wingspan. For forwards and centers, height and wingspan were also determinants, as were speed and agility [17]. In another study, Combine measurements and on-court performance of drafted players, revealed a positive association between anthropometric measurements and performance. This study makes groups with Combine variables, establishing three subscales of all measurements. Moreover, on-court performances are not individual statistics, are they are based on team performance [18]. For all the above, the purpose of this research was to identify the Combine tests that explain draft position and future performance in the NBA rookie season. The specific objectives were: (i) to predict what Combine measures explain draft position, in order to establish what tests are useful; (ii) to analyze what Combine measures are related to game performance; and (iii) to analyze draft position and rookie year performance.

## 2. Materials and Methods

### 2.1. Participants

The participants in the study were those basketball players that participated in the NBA Draft Combine from 1999 to 2018 (20 seasons). Data were retrieved from https://stats.nba.com/draft/combine/ [19], the official website of the Combine. Only drafted players (who subsequently played in the NBA) were selected for the study (n = 723). The Combine is composed of three different types of test: anthropometric measurements, strength and agility tests, non- stationary and spot up shooting tests. For the present study only anthropometric and strength and agility test were used. This is a retrospective study design, using data from Combine, which are available in various public domains. European General Data Protection Law procedures were followed in order to maintain the anonymity of the sampled players. This study was conducted as a secondary statistical analysis of data available through web-based public access with no individual health information.

### 2.2. Variables and Procedures

Variables were selected from the previous categories of anthropometric measurements and strength and agility tests. Anthropometric measures included: (i) Height without shoes (H): using a physician scale measured in feet and inches; (ii) Weight (W) (kg): measured with a physician scale; (iii) Wingspan (WS): Measured using a measuring tape in feet and inches, from the tip of the left hand to the tip of the right hand with the arms held horizontally; (iv) Standing Reach (SR), measured with a measuring tape, from the floor to the longest tips of the hands with both arms reaching up and players standing straight; (v) Body fat Percentage (BFAT), using a skinfold caliper, measuring the skinfold thickness of pectoral, abdomen, and quadriceps; (vi) Hand Length (HL) (cm), The length from the bottom of the player’s palm to the tip of the middle finger measured with a measuring tape; and (vii) Hand Width (HW, cm), the length from the tip of the player’s thumb to the tip of the little finger measured with a measuring tape. Strength and agility tests included: (i) Strength (Str): 83.9 kg (135 pounds) Bench Press, maximum number of bench press repetitions at 83.9 kg with proper technique is recorded; (ii) Power: Vertical Jump (VJ). The vertical jump is an athlete’s explosive leg power test. Two versions of this test are performed, the standard (no step) vertical jump and a running Max Vert (RVJ). (iv) Speed: ¾ Court Sprint (¾ CS), time to sprint over the distance of three quarters of the court is measured in seconds; (v) Agility: Lane Agility Drill (LAD), starting from the left corner of the free-throw line, with one foot behind the line, no rocking movement allowed. Run forwards to the baseline. At the cone, change movement to a side shuffle, and move sideways to the right across the baseline. At the next cone back pedal up the lane to the foul line, then side shuffle left back towards the start line. Here the subject touches the floor at a point even with the starting cone, then reverses direction to return back around the course to complete another revolution. First side shuffle right, forward sprint, side shuffle left then back pedal to complete the test. Remain facing forwards towards the baseline throughout the test. Two trials are allowed (Figure 1); (vi) Agility: Reactive Shuttle Run (RSR), the player starts in the middle of the key and runs to each side of the key before returning to the center (Figure 1).

Moreover, performance of the rookie year was measured through game related statistics: points scored, one-, two- and three-point field-goals attempted, converted and percentage, defensive and offensive rebounds, assists, turnovers, steals, blocks and fouls. Draft position was also collected from 1 to 60 (first and second round). Players were classified into backcourt players and frontcourt players, according to the information available at the moment in the draft (accessed in September 2019). Variables measured in inches and pounds were converted into cm and kg manually.

### 2.3. Statistical Analysis

First, a descriptive analysis with mean and standard deviation was carried out for different player positions, frontcourt and backcourt, in order to characterize the sample. In all the variables the number of players was noted, as not all players were evaluated in all variables. In second place, a regression analysis was carried out. Combine variables, anthropometric and agility/strength variables were analyzed to explore their effect on draft position. The model is as follows:DP = β1 + β2 × H + β3 × W + β4 × WS + β5 × SR + β6 × BFAT + β7 × HL + β8 × HW + β9 × Str + β10 × VJ + β11 × RVJ + β12 × ¾ CS + β13 × LAD + β14 × RSR + εi

The Durbin-Watson test was used to check whether the residuals in the model were independent and look within the data for controlling collinearity effects. Finally, a correlation analysis was performed to identify relationships among: (i) Combine anthropometric and strength and agility measures and game performance through game related statistics; and (ii) the draft position and game performance using Pearson′s correlation coefficients. Statistical significance of Pearson′s correlation coefficients depend on sample size, so effect sizes of correlations were reported because of varied sample size, and Cohen’s effect size criteria for correlation coefficients was used to interpret it (small: |r| = 0.10–0.29, medium: |r| = 0.30–0.49, and large: |r| = 0.50 [20]. The analyses were performed using IBM SPSS software v. 22 (IBM Corp., Armonk, NY, USA), and the significance level was set at *p* < 0.05.

## 3. Results

In the first place, descriptive statistics are shown in Table 1. Results show how frontcourt players achieved higher values in anthropometric variables, whereas backcourt players scored more than frontcourt players in strength and agility, with the exception of the bench press.

Table 2 presents the regression analysis results. Durbin-Watson values in both regressions are near to 2, meaning that data are independent. Only two variables in each regression achieved significant values. Hand width and height in frontcourt players, and standard vertical jump and running vertical jump in backcourt players. All β-values are negative, but standard vertical jump. Negative values mean that, if the variable increase, the draft position is better. Whereas the standard vertical jump increase, the draft position also increases.

Results for correlation analysis between game related statistics and combine test are presented in one table. Variables with correlation coefficients higher than |0.20| are shown. Table 3 shows the relationship in frontcourt and backcourt players. Whereas many correlations have been found, only two relationships are considered medium (<|0.30|), wingspan and blocks, and reactive shuttle run and draft position. Both coefficients are positive, meaning that the higher the value in wingspan and reactive shuttle run, the better the draft position and number of blocks. Moreover, in Table 3 correlations for backcourt players are shown, only the relationship between height and assists and blocks are medium. However, there are correlations between |0.24| and |0.29|. Height is positively related with offensive rebounds (0.28) and blocks (0.27) and negatively with assists (−0.27). Wingspan has the same relationship with offensive rebounds (0.26) and blocks (0.24) and a negative one with assists (−0.27).

Finally, Table 4 presents the relationship between draft position and game performance indicators. All game performance indicators are negatively correlated with draft position in all players, and frontcourt and backcourt players separately. These results indicated that the better the draft position (nearer to 1), the higher the scores in all game performance indicators. Points, Field-Goals made and Field-goals tried are strongly related to draft position (<|50|).

## 4. Discussion

Results show that the Combine tests, with the exception of hand width and height in frontcourt players and standard vertical jump and running vertical jump in backcourt players, do not predict draft position. Future performance indicators were explained by several Combine tests in all players. Specifically, stronger relationships were wingspan with blocks and shuttle run with draft in frontcourt players; and between height with assists and blocks in backcourt players. In recent years, few studies have examined the validity of the NBA Draft Combine to identify and select young talented players [17,18]. To the best of our knowledge, this is the first study to examine combine measures with performance indicators in depth, analyzing all data raw. The purpose of this investigation was to identify the Combine tests that explain draft position and future performance in the NBA rookie season.

Some studies have accounted for the importance of accurate assessment of players for NBA teams. This is crucial to make the best draft election [18] or be conscious of specific demands of different players’ positions [17]. In addition, as the NBA is the most important basketball competition in the world, the findings offer teams and organizations a deeper understanding of talent identification for future performance [1].

Some studies have attempted to identify and classify the efficacy of different Draft Combines, (NBA, NFL and NHL) in predicting future performance in the sport. The NFL Combine has identified those tests and measures that successfully predict performance in different playing positions [1]. These identifications were in the positions that are most dependent on speed and agility, which is what Combine assesses. In fact, the authors stated that the Combine does not measure those factors that are important for success. In the NBA Combine, differences in anthropometric characteristics and physical fitness ability were found between drafted and non-drafted players [17]. Also, specific measures were useful for being selected as guards (leg power) and power forwards and centers (agility and speed).

One study in basketball stated that the Draft Combine has value for predicting future players′ performance [18]. This study reduces all measures to three components: length-size, power-quickness, and upper-body strength. Also, on-court performance was a calculation of different metrics (Player Efficiency Rating, Offensive Win Shares, Defensive Win Shares, Win Shares, Win Shares Per 48 min, Offensive Box Plus/Minus, Defensive Box Plus/Minus, Box Plus/Minus, and Value Over Replacement Player) and almost all are based on team performance. This study finds that the Draft Combine predicts future performance in the first year and three years when variables were grouped.

It has been stated, that beyond anthropometric and fitness variables, there are other factors that have an important prediction power on future performance. Mulholand and Jensen [21] described “football intelligence” as the ability to learn offensive schemes, memorize routes or recognize patterns in the defense to attack or catch passes. McGee et al. [1], establish that quickness to react and the ability to read the offense is fundamental in football, as well as an athlete’s determination, toughness, and ability to work as part of a team. In basketball players, these variables are even more important than in football. Players are in continuous interaction with teammates and opposing players in chaotic interactions and unexpected contexts [22]. None of the Combine measures challenge players to find contextual information and choose the accurate solution for each game situation [23,24]. The NBA Draft Combine performance measurements only measure physical capabilities.

In fact, some of the measurements of the Combine are correlated with each other. Power measurements are highly correlated with speed assessment [1,16]. According to this information, it seems that lane agility drill, reactive shuttle run, ¾ court sprint, standard vertical jump and running vertical jump are measuring the same ability. In this respect, the Combine measurements are not sport specific, they are general and common tests used to assess physical condition in basketball [25]. It is necessary to choose specific tests for each sport modality and position that are related to the sport and report specific results with greater validity [26,27,28].

Other studies have attempted to identify the key components and variables that players and coaches have to development to reach a high performance in basketball [29]. This study collects information from professional coaches and international basketball players (among other experts) and states that it is the interaction of a multitude of social, biological and psychological factors that lead to best performance. In fact, other studies have identified 12 factors that contribute to the success of Olympic champions in the U.S. [30,31]. Saénz-López et al., [29] pointed out that the environmental factors are the most important to reach high level performance, followed by psychological, tactical and technical factors. Finally, this sample of experts identified anthropometry and physical condition, as the two factors that the Combine tries to measure.

Some of the reasons for NBA teams to not choose players according to the results of these test could be: (i) the belief that these physical skills can be taught; (ii) the training effect: there are players that prepare these specific tasks in order to perform better on the test day [1]. On the other hand, on-field performance in college is likely the strongest predictor of success in the NFL [32]. In basketball, similar findings have been presented. Scoring performances in the NCAA tournament are related to draft placement and the probability of being drafted [33]. So, the question is why is the Combine anthropometric and power-agility test conducted in its current state?

## 5. Conclusions

From the results of the current study, it is suggested that battery tests have to be modified, including measures that are sport specific. In addition, other attributes and characteristics of players, such as psychological factors or mental strength, have to be measures along with college on-field performance indicators. The Draft Combine costs a lot of money, and teams send scouts and coaches to the event in order to get better knowledge of the possibly drafted players. Moreover, physical condition seems not to be the key factor to advance to the next level. Tactical and technical level or on field performance receive much attention during the NCAA season. It could be important to measure other factors in the Combine that are not as evident as those, such as mental preparation, game intelligence, and others for which tools for measurement are available. As practical applications, other sports should design their own Combine tests according to game demands and specifications of different player positions and functions. In basketball, teams and technical staff have to eliminate all the paraphernalia of the Combine, and focus on what is important for their organizations, along with on-court performance.

## 6. Limitations

There are limitations to this study to take into consideration. It is a secondary analysis of data not directly collected by the authors so it was impossible to test validity measures and collection technique. The correlations model is presented with a large number of relationships, and several spurious findings are shown, but only those with medium effects are commented on. The existence of a high number of significant but spurious correlations could explain the little importance given to the measurements in the NBA teams draft elections. However, the study has used data available from 20 years, strengthening the validity of the present findings.

## 7. Future Research

In future research other variables have to be included, such as the Combine shooting test, and advanced statistics (e.g., offensive rating, defensive rating, net rating, assist to turnover ratio, effective field goal percentage, player impact estimate, and more).

## Figures and Tables

**Figure 1 ijerph-17-07355-f001:**
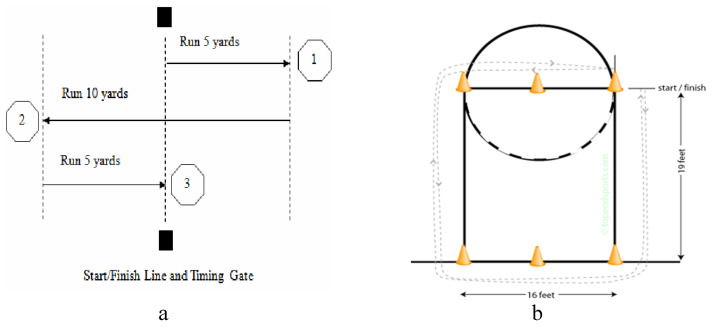
Lane Agility Drill (**a**) and Reactive Shuttle Run test description (**b**).

**Table 1 ijerph-17-07355-t001:** Descriptive statistics (M and SD) of the NBA Combine according to player’s position.

Combine Measures	Frontcourt			Backcourt		
	*N*	*M*	*SD*	*N*	*M*	*SD*
Body fat	222	8.51	3.21	403	6.54	1.83
Hand Length	136	9.08	0.38	245	8.56	0.38
Hand Width	136	9.71	0.71	245	9.20	0.59
Height	262	2.08	0.04	457	1.94	0.07
Weight	262	108.76	8.51	456	91.59	7.82
Wingspan	263	2.20	0.06	457	2.08	0.12
Lane Agility Drill	223	11.67	0.56	406	11.15	0.67
Reactive Shuttle Run	58	3.11	0.18	131	3.06	0.16
¾ Court Sprint	225	3.35	0.13	409	3.23	0.10
Standard vertical jump	227	28.67	2.87	411	30.14	2.87
Running Max Vert	227	32.98	3.07	410	35.87	3.38
Bench Press	198	12.03	5.19	345	9.73	4.81

**Table 2 ijerph-17-07355-t002:** Effects of Combine test on overall draft position according to playing positions.

Combine Measures	Frontcourt					Backcourt				
	Non- Standardized Coefficients		Standardized Coefficients			Non- Standardized Coefficients		Standardized Coefficients		
	B	Standard error	Beta	t	Sig.	B	Standard error	Beta	t	Sig.
(Constant)	568.85	227.87		2,49	0.02	28.79	110,60		0.26	0.79
Body fat	−0.89	1.29	−0.16	−0.69	0.49	0.05	1.31	0.00	0.04	0.96
Hand Length	−4.28	8.37	−0.11	−0.51	0.61	0.16	6.08	0.00	0.02	0.97
Hand Width	−8.71	4.46	**−0.40**	−1.95	**0.06**	1.18	3.64	0.04	0.32	0.74
Height	−170.25	79.064	**−0.49**	−2.13	**0.04**	−19.00	41.67	−0.08	−0.45	0.65
Weight	0.27	0.54	0.13	0.51	0.61	0.32	0.45	0.14	0.71	0.47
Wingspan	43.04	46.30	0.22	0.93	0.36	−26.96	23.71	−0.17	−1.13	0.25
Lane Agility Drill	−1.33	6.43	−0.05	−0.20	0.83	0.19	1.48	0.01	0.13	0.89
Reactive Shuttle Run	5.89	18.89	0.08	0.31	0.75	16.33	10.65	0.18	1.53	0.13
¾ Court Sprint	−19.03	26.55	−0.17	−0.71	0.48	7.72	21.00	0.05	0.36	0.71
Standard vertical jump	−1.18	1.67	−0.25	−0.70	0.48	2.36	1.25	**0.43**	1.89	**0.06**
Running Max Vert	−2.40	1.81	−0.54	−1.32	0.20	−2.64	1.00	**−0.59**	−2.64	**0.01**
Bench Press	0.36	0.67	0.11	0.54	0.59	0.22	0.42	0.07	0.51	0.60

Dependent variable: draft. In **bold**: *p* > 0.05.

**Table 3 ijerph-17-07355-t003:** Pearson correlation coefficients between COMBINE measures and game performance indicators in Frontcourt and Backcourt players (r value and (n)).

Frontcourt	Draft	Points	FG Made	FG Tried	3PFG Made	3PFG Tried	FTm	FT Tried	OFF REB	DEF REB	Assists	TO	STLs	BLK	Fouls Committed	Fouls Received
Wingspan	**−0.137 * (263)**	−0.01 (243)	−0.01 (243)	−0.04 (243)	**−0.153 * (243)**	**−0.163 * (243)**	0.03 (243)	0.05 (243)	0.09 (243)	0.05 (243)	−0.10 (243)	0.07 (243)	0.01 (243)	**0.368 ** (243)**	0.09 (243)	0.06 (243)
Reactive Shuttle Run	**0.316 * (58)**	−0.10 (54)	−0.08 (54)	−0.08 (54)	0.00 (54)	−0.01 (54)	−0.16 (54)	−0.14 (54)	−0.06 (54)	−0.18 (54)	−0.26 (54)	−0.310 * (54)	−0.19 (54)	−0.10 (54)	−0.25 (54)	−0.14 (54)
**Backcourt**																
Hand Length	**−0.126 * (245)**	−0.10 (230)	−0.09 (230)	−0.11 (230)	−0.10 (230)	−0.10 (230)	−0.09 (230)	−0.07 (230)	**0.178 ** (230)**	0.12 (230)	**−0.267 ** (230)**	**−0.156 * (230)**	0.02 (230)	**0.182 ** (230)**	0.05 (230)	−0.06 (230)
Height	−0.07 (457)	**−0.123 * (418)**	**−0.112 * (418)**	**−0.135 ** (418)**	**−0.139 ** (418)**	**−0.146 ** (418)**	**−0.117 * (418)**	−0.09 (418)	**0.237 ** (418)**	**0.120 * (418)**	**−0.416 ** (418)**	**−0.263 ** (418)**	**−0.148 ** (418)**	**0.293 ** (418)**	−0.05 (418)	**−0.124 * (418)**
Weight	−0.105 * (456)	0.01 (417)	0.01 (417)	−0.01 (417)	−0.05 (417)	−0.05 (417)	0.04 (417)	0.07 (417)	**0.280 ** (417)**	**0.214 ** (417)**	**−0.270 ** (417)**	**−0.128 ** (417)**	−0.03 (417)	**0.269 ** (417)**	0.07 (417)	0.01 (417)
Wingspan	−0.07 (457)	−0.05 (418)	−0.04 (418)	−0.06 (418)	**−0.114 * (418)**	**−0.105 * (418)**	−0.01 (418)	0.02 (418)	**0.259 ** (418)**	**0.135 ** (418)**	**−0.269 ** (418)**	**−0.127 ** (418)**	−0.04 (418)	**0.237 ** (418)**	0.02 (418)	−0.04 (418)
Running Max Vert	−0.10 (410)	**0.101 * (374)**	0.10 (374)	0.09 (374)	0.03 (374)	0.05 (374)	**0.123 * (374)**	**0.121 * (374)**	0.00 (374)	0.08 (374)	0.07 (374)	0.09 (374)	0.07 (374)	0.03 (374)	0.06 (374)	0.09 (374)

* *p* > 0.05; ** *p* > 0.01; **In bold**: *p* > 0.05 for a better identification.

**Table 4 ijerph-17-07355-t004:** Pearson correlation coefficients between draft position and game performance indicators in All, Frontcourt and Backcourt players.

Game Performance Indicators	All (n = 663)	Frontcourt (n = 244)	Backcourt (n = 419)
	Draft Position	Draft Position	Draft Position
Points	−0.513 **	−0.553 **	−0.516 **
Field-Goals made	−0.514 **	−0.543 **	−0.515 **
Field- Goals tried	−0.504 **	−0.564 **	−0.514 **
3-Point field- goals made	−0.216 **	−0.145 *	−0.322 **
3-Point field- goals tried	−0.209 **	−0.118	−0.328 **
Free Throws made	−0.480 **	−0.510 **	−0.470 **
Free Throws tried	−0.488 **	−0.511 **	−0.473 **
OFF REB	−0.365 **	−0.493 **	−0.317 **
DEF REB	−0.495 **	−0.560 **	−0.460 **
Assists	−0.282 **	−0.396 **	−0.348 **
Turnovers	−0.471 **	−0.552 **	−0.477 **
Steals	−0.390 **	−0.432 **	−0.413 **
Blocks	−0.354 **	−0.471 **	−0.299 **
Fouls committed	−0.478 **	−0.522 **	−0.449 **
Fouls received	−0.527 **	−0.568 **	−0.509 **

* *p* > 0.05; ** *p* > 0.01.

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
