# Peer review of "Citius, Altius, Fortius; Is It Enough to Achieve Success in Basketball?"

_ijerph, 2020, doi:10.3390/ijerph17207355_

Round 1

Reviewer 1 Report

Thank you for submitting your article titled "Citius, Altius, Fortius??? Are really necessary to success at basketball?" to IJERPH. 

I hope my comments will help strengthen the paper. My major comments include the following:

- Please ensure a native English speaker rereads the paper in its' entirety and correctly edits for grammar and proper sentence structure.

Title: Not quite sure this makes sense - "Are really necessary to success at basketball?" - consider rechecking English grammar

Abstract: Please read over the entirety of the abstract and edit for grammar. E.g., line 12 - "included" should this read includes? Why is combine and draft in all capitals on line 14 and not elsewhere? Another example includes: "Moreover, correlation analysis was performed" - I think this should be a correlation analysis was performed (if it was one) or correlation analyses were performed if more than one?

Introduction: there is a track changes comment included that should be deleted (line 40). The introduction also begins with "Error! Bookmark not defined".

line 28 - perhaps give the reader examples of these physical/athletic capacities

line 31 - define NBA and NFL

line 32 - "always have productive careers" - successful careers? 

lines 32-34 - "Moreover, research has pointed out that best team recruit young players according to their chronological age [7]. Meaning that more mature players have better opportunities of being scouted." - need major English editing 

It is very difficult to understand the majority of the introduction. It needs substantial editing. 

e.g., line 56 - "stablishing" - establishing?

Methods: revise/edit for quality of writing 

Results: consider condensing so you do not have as many tables. It is very difficult and time consuming for the reader to sift through so much data in the tables. 

Discussion and conclusion: very difficult to understand and see any logistical flow. Should the limitations come before the conclusion?  

Author Response

Thank you for submitting your article titled "Citius, Altius, Fortius??? Are really necessary to success at basketball?" to IJERPH. I hope my comments will help strengthen the paper. My major comments include the following:

- Please ensure a native English speaker rereads the paper in its' entirety and correctly edits for grammar and proper sentence structure.

Title: Not quite sure this makes sense - "Are really necessary to success at basketball?" - consider rechecking English grammar.

Title have been changed: “Citius, Altius, Fortius; Citius, Altius, Fortius; Is enough to success at basketball?

Abstract: Please read over the entirety of the abstract and edit for grammar. E.g., line 12 - "included" should this read includes? Why is combine and draft in all capitals on line 14 and not elsewhere? Another example includes: "Moreover, correlation analysis was performed" - I think this should be a correlation analysis was performed (if it was one) or correlation analyses were performed if more than one?

Ok, all the suggestions has been attended.

Introduction: there is a track changes comment included that should be deleted (line 40). The introduction also begins with "Error! Bookmark not defined".

Ok, removed.

line 28 - perhaps give the reader examples of these physical/athletic capacities

Thanks for the suggestions. Done.

line 31 - define NBA and NFL

Thanks, done

line 32 - "always have productive careers" - successful careers? 

Ok, corrected

lines 32-34 - "Moreover, research has pointed out that best team recruit young players according to their chronological age [7]. Meaning that more mature players have better opportunities of being scouted." - need major English editing 

It is very difficult to understand the majority of the introduction. It needs substantial editing. 

Thanks for the suggestions. The manuscript have been revised by a native English speaker.

e.g., line 56 - "stablishing" - establishing?

Ok, done.

Methods: revise/edit for quality of writing 

Thanks for the suggestions. The manuscript have been revised by a native English speaker.

Results: consider condensing so you do not have as many tables. It is very difficult and time consuming for the reader to sift through so much data in the tables. 

Table 3 and 4 have been collapsed. Only substancial correlations have been shows.

Discussion and conclusion: very difficult to understand and see any logistical flow. Should the limitations come before the conclusion?  

Reviewer 2 Report

The manuscript was prepared very well. The introduction section justifies the purpose of the study. I recommend this work for publication and would like to congratulate the authors for their excellent study. However, I have the following comments:

Introduction

Page 1 Line 31

Please describe the NBA and NFL acronyms and indicate which sports and country they correspond to

Page 1 Lines 34-36

Authors: “In this sense, it has been hypothesized that, perhaps, on-court performance is a better indicator of future performance than fitness measurements in Combine”.

What does it mean? You have formulated the hypothesis? what is the performance on the court? and what are the "Combine" measures? Please rewrite this part, trying to give a clear explanation to the reader.

Page 1 Lines 36-38

There are a number of on-court performance metrics available and new player selection systems that allow for different combinations of players and on-court performance variables.

Please insert a table indicating which are the varying on-court performance metrics. Explain what the combinations of players and on-court performance variables are.

Page 2 Lines 45-56

This paragraph describes, measures and skills related to basketball correctly. However, comparing them to 2 different sports (football and hockey), please explain more precisely the similarities they may have.

Page 2 Lines 56-59

Explain more precisely the general objective and secondary objectives of the study.

Methods

The methodology is correct and very well structured.

Results

The results are well described and do not require any suggestions. However, it would be necessary to improve the legends of the figures for a better understanding. For example, tables 3 and 4.

Discussion

  • The Discussion section includes the accurate reference of the results obtained to the studies of other authors. However, it is necessary to include initial paragraph explaining the most relevant and novel findings of the study.
  • What does the NCAA stand for?
  • The system that the authors design in the study could be applied to other sports disciplines? A paragraph of potential practical applications could be included
  • The authors have included a section of limitations that describes exactly and precisely. However, I recommend adding the strengths of the study to this section.

Author Response

Comments and Suggestions for Authors

The manuscript was prepared very well. The introduction section justifies the purpose of the study. I recommend this work for publication and would like to congratulate the authors for their excellent study. However, I have the following comments:

Introduction

Page 1 Line 31

Please describe the NBA and NFL acronyms and indicate which sports and country they correspond to

Ok, Done.

Page 1 Lines 34-36

Authors: “In this sense, it has been hypothesized that, perhaps, on-court performance is a better indicator of future performance than fitness measurements in Combine”.

What does it mean? You have formulated the hypothesis? what is the performance on the court? and what are the "Combine" measures? Please rewrite this part, trying to give a clear explanation to the reader.

Ok, thanks for the suggestions. The text have been rewritten as follows:

“In this sense, it has been hypothesized that, perhaps, on-court performance, measured from game-related statistics (points, rebounds,..), is a better indicator of future performance than fitness measurements in Combine, measured with physical test (anthropometric and strength and agility)[8]”

Page 1 Lines 36-38

There are a number of on-court performance metrics available and new player selection systems that allow for different combinations of players and on-court performance variables.

Please insert a table indicating which are the varying on-court performance metrics. Explain what the combinations of players and on-court performance variables are.

Thanks for the comment. We think that include a new table with information that is not relevant for the study, do not help to the reader to understand the problem. We have decide to erased this part of the text.

Page 2 Lines 45-56

This paragraph describes, measures and skills related to basketball correctly. However, comparing them to 2 different sports (football and hockey), please explain more precisely the similarities they may have.

Ok, thanks for the suggestion. The text have included: “On those sports physical contact are allow and necessary to succeed; speed is fundamental for some playing positions in football or hockey. In basketball, the playing field is smaller, so speed is not as crucial as in football. However, in both sports context is continuously changing, being unpredictable, so others skills are so important to succeed”

Page 2 Lines 56-59

Explain more precisely the general objective and secondary objectives of the study.

Ok, The following text have been included: “Specifically, the objectives of this research were: i) to predict what Combine measures explain draft position, in order to stablish what tests are useful; ii) to analyze what Combine measures are related with game performance; and iii) to analyze Draft position and rookie year performance.”

Methods

The methodology is correct and very well structured.

Thanks for the comment

Results

The results are well described and do not require any suggestions. However, it would be necessary to improve the legends of the figures for a better understanding. For example, tables 3 and 4.

Tables 3 and 4 have been merged in one table according to other reviewer suggestions.

Discussion

The Discussion section includes the accurate reference of the results obtained to the studies of other authors. However, it is necessary to include initial paragraph explaining the most relevant and novel findings of the study.

Thanks for the suggestion. In the first paragraph of the Discussion the following text is included:

“Results show that Combine test do not predict DRAFT position, with exception of hand width and height in frontcourt players, and standard vertical jump and running vertical jump. Future performance indicators were explained by several COMBINE tests in all players. Specially, stronger relationships were wingspan with blocks and shuttle run with draft in frontcourt players; and between height with assists and blocks in backcourt players.”

This text have been changes to the beginning of the paragraph.

What does the NCAA stand for?

We do not take into consideration what NCAA stands for. We analyze the Combine in its current form.

The system that the authors design in the study could be applied to other sports disciplines? A paragraph of potential practical applications could be included.

Ok, thanks for the suggestion. The following text have been included:

“As practical applications, others sports should design their own Combine test according to game demands and specifications of different player positions and functions. In basketball, teams and technical staff have to eliminate all the noise of the Combine, and focus on what is important for their organizations, along with on-court performance.”

The authors have included a section of limitations that describes exactly and precisely. However, I recommend adding the strengths of the study to this section.

Ok, included.

Reviewer 3 Report

The authors wrote an interesting manuscript; it deals with the association between draft combine measures, draft position and game-related statistics. The theoretical background is relevant to the topic and provides general information. In line no.30, after the citation “[2, 3]” the full stop is missing. Line no.46 includes the phrase “athletics measurement,” but in line no.30, the authors used the phrase “physical fitness assessment.” Please, use the same terminology through the whole text. A comma is missing in line no.48, after the word “strength.” In the same line, (no.48) is the phrase “physical size.” The authors should define what the physical size is. Is it body height, body weight, or what? In line no.51, after the word “positions” the full stop is missing.

Part “Material and Methods” is understandable. There is only one thing that requires an explanation. In line no.100 is stated that authors used for analysis variables “game played” and “minutes.” In results, the correlation between the combine variables, the draft position and “games played” and “minutes is not mentioned.” The authors should explain.

“Results” are presented in a well-arranged way. “Discussion” is apt, and “Conclusion” is comprehensible. However, there are some formal shortcomings.

Throughout the whole text, sometimes authors used the capitalized words “COMBINE” and “DRAFT”, and sometimes is capitalized only the first letter “Combine” and “Draft.” Please, unify it.

I appreciate part “Limitations,” where the authors are aware of the limitations of the study. However, authors should also state that for future research in this area the combine shooting test, and advanced statistics (e.g. offensive rating, defensive rating, net rating, assist to turnover ratio, effective field goal percentage, player impact estimate, and more) should be included.

Author Response

Comments and Suggestions for Authors

The authors wrote an interesting manuscript; it deals with the association between draft combine measures, draft position and game-related statistics. The theoretical background is relevant to the topic and provides general information. In line no.30, after the citation “[2, 3]” the full stop is missing. Line no.46 includes the phrase “athletics measurement,” but in line no.30, the authors used the phrase “physical fitness assessment.” Please, use the same terminology through the whole text. A comma is missing in line no.48, after the word “strength.” In the same line, (no.48) is the phrase “physical size.” The authors should define what the physical size is. Is it body height, body weight, or what? In line no.51, after the word “positions” the full stop is missing.

Ok, all the suggestions have been attended. Physical size has been change for body composition, that is accurate.

Part “Material and Methods” is understandable. There is only one thing that requires an explanation. In line no.100 is stated that authors used for analysis variables “game played” and “minutes.” In results, the correlation between the combine variables, the draft position and “games played” and “minutes is not mentioned.” The authors should explain.

Thanks for the suggestions. These variables were used and then deleted. Corrected

“Results” are presented in a well-arranged way. “Discussion” is apt, and “Conclusion” is comprehensible. However, there are some formal shortcomings.

Thanks for the comments.

Throughout the whole text, sometimes authors used the capitalized words “COMBINE” and “DRAFT”, and sometimes is capitalized only the first letter “Combine” and “Draft.” Please, unify it.

Ok. Corrected.

I appreciate part “Limitations,” where the authors are aware of the limitations of the study. However, authors should also state that for future research in this area the combine shooting test, and advanced statistics (e.g. offensive rating, defensive rating, net rating, assist to turnover ratio, effective field goal percentage, player impact estimate, and more) should be included.

Ok, Included.

This paragraph have been included:

“7. Future Research

In future researches other variables have to be included, such as combine shooting test, and advanced statistics (e.g. offensive rating, defensive rating, net rating, assist to turnover ratio, effective field goal percentage, player impact estimate, and more).”

Round 2

Reviewer 1 Report

Thank you for making revisions to the manuscript. Upon further reading, I am sticking with my decision that the paper should be rejected from this journal. Firstly, it is clear the quality of English has not been improved. For example, the title reads "Citius, Altius, Fortius; Is enough to success at basketball?" - this doesn't make sense and is grammatically incorrect. I begin with the title, there are many more errors in the abstract and entire paper, I could go on and on. In addition to this, I feel strongly about the suitability of this paper to the International Journal of Environmental Research and Public Health, a high-quality and peer reviewed journal. Frankly, I feel the paper is not entirely suited to this journal and should be resubmitted elsewhere for more apt suitability. Finally, in my opinion this research does add a significant amount to the field. I mean for these comments not to disappoint, but rather improve resubmission to a more suitable journal. 

Author Response

Thanks for the suggestions, The manuscript has been thoroughly reviewed

Reviewer 2 Report

The authors with their reviews have satisfied all the corrections that were proposed. 

Author Response

thanks,